# Comfort and Time-Based Walkability Index Design: A GIS-Based Proposal

**DOI:** 10.3390/ijerph16162850

**Published:** 2019-08-09

**Authors:** Tarek Al Shammas, Francisco Escobar

**Affiliations:** Department of Geology, Geography and Environmental Sciences, University of Alcalá, Calle Colegios 2, Alcalá de Henares, 28801 Madrid, Spain

**Keywords:** walkability index, comfort, weighting, secondary data sources, GIS

## Abstract

Encouraging people to walk as a means of transport throughout their daily lives has obvious benefits for the environment, the economy, and personal health. Specific features of the built environment have a significant influence on encouraging or discouraging walking. By identifying and quantifying these features we can design Walkability Indices (WI). The WI in the literature do not take factors related to comfort such as noise pollution and shade/sun conditions into account. Given the importance of these factors in walking, we decided to include them in our design of a new geographic information system (GIS)-based WI. The relative weight of each factor was determined by consulting experts. The proposed WI, computed for the entire city of Madrid, Spain, uses sections of the sidewalk as the spatial unit. The properties of this WI (based on secondary sources, spatially detailed, dynamic, weighted, and including comfort-related factors) fill a gap in previous WI proposals.

## 1. Introduction

There is wide evidence [1,2] to show that when the urban population takes up walking as a daily activity there is a positive impact on health [3,4], transportation [5,6], economy [7], and air quality [8]. Walking can be promoted as a regular activity [9] and as a means of transport [10,11] when specific infrastructure features are available [7,12,13].

A considerable amount of research has been carried out to examine which aspects of the infrastructure encourage people to walk [12,14,15]. In previous research, the following built environment features were classified as factors affecting walking: Functionality, safety and aesthetics of sidewalks, population density, the proximity of destinations, and street connectivity [14,15,16,17,18,19,20]. These factors were then applied in different combinations in the design of an index measuring the walkability of the city in question [21,22,23,24,25].

Previous studies [25,26] highlighted the importance of factors related to walkers’ comfort such as noise and shade but did not explicitly compute them in a specific walkability index (WI). The other available WI [27,28] did not take any comfort-related factors into account, probably because they were not considered important or due to a lack of data and computing capabilities.

Past WI proposals not only differed in terms of the factors they considered but also in terms of the data collection methods and the level of detail [25,27]. A number of studies have relied on field surveys or questionnaire-based data, but this approach requires a huge investment in time and resource to obtain detailed results. Other studies, by contrast, preferred to sacrifice the level of detail in favor of a more affordable WI based on secondary data [29,30]. In the latest WIs, the unit of observation is larger, implying lower spatial precision. This is the case of Walk Score [31], which was developed with available secondary data. In view of the difficulties involved in taking all the relevant variables into account, the same authors later updated this score with additional built environment measures to provide a more precise WI [32]. Table 1 lists the factors and the data type used to calculate the WI in different countries. The information included in this table does not intend to be exhaustive but a selection of the main WI types found in the literature. As shown in Table 1, the majority of WIs rely on secondary data and use three factors (population density, activities, and connectivity). Additional factors, such as cleanliness, aesthetics, and safety have been considered in some WI but their computation required intensive fieldwork, thus it has been computed only in small areas.

An additional problem is that currently available WI have largely overlooked the importance of the relative weighting of the factors included in the index [15,42]. For example, Gullón et al. [34] weighted all the factors equally. Frank et al. [21] considered that street connectivity should be given double the weight of the other factors, which were all weighted equally. Other studies have acknowledged the importance of weighting the three traditional factors of walkability (street connectivity, residential density, and land-use mix) differently when assessing transport walking [43]. Furthermore, in a study supported by an online survey, researchers assigned relative weights to each of the factors considered for the Walkability Index [42]. These varying approaches show that the debate regarding the weighting of the various factors remains open, especially since new factors for the Walkability Index are continually being proposed.

Current geographic information system (GIS) developments [44] and the availability of geospatial data [45] make it easier to quantify the different aspects of our cities that encourage walking. These aspects can be combined to form a walkability index [46,47].

The main objective of this study is to propose a new WI, which will have the following advantages compared to existing WI:All required data are readily available. It is possible to compute the proposed WI without the need for field work.The proposed WI includes comfort factors (noise and sun/shade conditions) that encourage or discourage walking. If shade is considered as a comfort factor, then the WI could be computed at different times during the day and on different days during the year. This makes the proposed WI dynamic and variable in line with local conditions.All factors involved are weighted according to their relevance for walking.

The availability of a dynamic WI that is easily computed under different conditions will help researchers, planners, and practitioners to understand the walking behavior of their fellow citizens and has the potential to support well-informed decisions to encourage walking.

## 2. Materials and Methods

### 2.1. Study Area

The study area for this research was the city of Madrid. Madrid is located in the center of Spain and enjoys a Mediterranean climate with continental influences [48]. Due to its altitude and dry climate, the weather can be extremely cold in winter and hot in summer with clear-sky conditions most days of the year [49]. These variations in weather conditions over the course of the year highlight the importance of taking into account the effects of shade/sun on walkability. High noise levels from urban traffic have been identified as a source of discomfort in many European capitals, including Madrid [50,51]. Despite this, the city of Madrid is generally flat, making walking relatively easy. The data required for the proposed index is also readily available in Madrid, making this city an ideal place to compute and test it.

### 2.2. Data Collection

Madrid City Council offers open access to data to be used in new research and applications [52]. This information was collected from various official institutions in Spain and the European Union [53,54,55], and most of it was updated annually. Table 2 summarizes the datasets used in this research. The height of buildings was available from the national Land Registry (https://www.sedecatastro.gob.es/).

Each of these datasets was provided in a different format and level of spatial aggregation (districts, census sections, sidewalks, street network axes, and buildings). This means that the first step was to integrate the data.

2016 was the year chosen for this study, as this was the most recent year with sufficient data.

### 2.3. Observation Units

Walkability factors were extracted from five aspects of the built environment of the city of Madrid. Three of them (population density, diversity of business activities, and connectivity) had already been presented in previous WI [40,43,56]. Two new factors (noise and sun/shade) were included in our proposed WI. Population density, diversity of activities, and connectivity factors were aggregated at the census section level, the smallest administrative unit in the Spanish Census. We digitized a noise level map, which was made available in hard copy by the City Council and was produced by weighting long-term average sound levels, determined for whole-day periods (from 7 to 7 pm) of the month. Polygons derived from this map were the result of interpolation procedures among noise observation stations. The sun/shade factor was obtained from extruded polygons representing buildings whose shadows were computed at different times.

### 2.4. Techniques

The proposed WI is dynamic and can be computed at any time of the day and on any day of the year. For demonstration purposes, in this paper, we computed the proposed WI on the summer and winter solstice at 2 particular times of each day. We decided to take one reading in the morning and one in the afternoon thus as to calculate changes in the WI due to the different positions of the shadows cast by the buildings. The summer solstice fell on a hot summer’s day, in which people prefer walking in shady areas rather than in the sun, and the opposite was true on the cold winter solstice.

### 2.5. Preparing the Factors

Since the datasets for the various walkability factors were provided in different formats, harmonization was needed prior to calculating the WI. Figure 1 shows the cartographic model we designed.

Population density (F1), diversity of activities (F2), and connectivity (F3) factors were described in each census section by the number of inhabitants per square kilometer, the number of different business activities per square kilometer, and the number of intersections in the street network with respect to its total length, respectively. Madrid City Council classified these activities into 19 different types (from agricultural to building, education and so on). What encourages walking is the diversity of activities; this is the number of different activities at a particular location, regardless of the type of activities concerned. Table 1 shows a number of WI found in the literature that adopted this factor for that reason.

The noise factor (F4) was extracted from a noise map [57], published in 2016. Ranges of sound levels (in dB) were assigned to the corresponding polygons. According to classifications of sound level comfort, 40 dB or less was considered pleasantly quiet, while 80 dB or above was considered disturbingly loud [58,59,60]. With this in mind, polygons with a sound level of ≤ 40 dB were assigned a value of F4 = 100, and those with a sound level of ≥80 dB were assigned a value of F4 = 0. Intermediate values were calculated by linear interpolation.

The shade factor (F5) was materialized as a set of polygons extracted from a 3D buildings model using the shadow volume tool in ArcScene [61]. This factor was binary and expressed the presence or absence of shade, F5 = {1,0} where Factor 1 was assigned to shade in summer and sun in winter. This factor was calculated for each of the particular cases (day of the year, time of the day) considered.

Once all the factors have been computed, they were then integrated at the sidewalk section level. After that, and given the differences between the units of the measured factors, their values must be normalized. The chosen range was [0–100] for population density (F1), diversity of activities (F2), connectivity (F3), noise (F4), and shade (F5) factors through the Equation (1).
(1)$$\mathrm{NF}[x]=\frac{F[x]-\mathrm{Min}\;(F[x])}{\mathrm{Max}\;(F[x])-\mathrm{Min}\;(F[x])}\cdot 100\;:\;x=(1,\;2,\;3,\;4,\;5)$$

### 2.6. Defining the Weights of the Factors

When adding new factors, it was important to assess the relative importance (relative weight) of each one. To this end, a questionnaire was prepared and sent to experts with experience in WI design from various relevant disciplines. The questionnaire asked subjects to express their opinions regarding the importance of population density, diversity of activities, connectivity, noise, shade in summer, and sun in winter, considering each factor individually (Appendix A). A study [62] reviewing research published between 1977 and 2015 in journals with impact factors of 1.5 or above was used to define the list of experts to whom the questionnaire was to be sent. Teaching and research staff and PhD students from the Department of Geology, Geography and Environmental Sciences at the University of Alcalá [63] and the Heart Healthy Hoods (HHH) Project [64] with experience in related topics were also contacted. In total, we sent 266 questionnaire invitations and obtained 66 responses; 3 from the University of Alcalá group, 5 from the HHH project and 58 from the list of experts extracted from the literature review [62]. Each response included an assessment on each of the factor’s importance in range (0–10). A mean of the importance of each factor was calculated as shown in Table 3. The importance of shade in summer and sun in winter were similar (with 5% difference). Then, by using means of factors’ importance, relatively factors’ weights were defined for all considered factors as shown in Table 3. Gathering population density weight (WF1), diversity of activities weight (WF2), connectivity weight (WF3), absence of Noise weight (WF4), and shade weight (WF5) equaled one.

### 2.7. Integration of Factors

Until this stage, all factors were presented with their normalized values in shapefile format polygons. All factors were presented in polygon features (census sections, noise, and shade polygons). We overlapped all factors polygons over the sidewalk polygons. We obtained split sidewalks based on the values of factors, on either side of the street. We calculated the WI for each of these parts of sidewalks by applying the Equation (2):WI = (WF1 ∗ NF1) + (WF2 ∗ NF2) + (WF3 ∗ NF3) + (WF4 ∗ NF4) + (WF5 ∗ NF5)(2)
where: WI is the walkability index, (NF1, WF1), (NF2, WF2), (NF3, WF3), (NF4, WF4), (NF5, WF5) are normalized and weighted population density, diversity of activities, connectivity, noise, and shade factors, respectively.

### 2.8. Tools

Maps were produced with the Geographic Information System (GIS) software ArcGIS (ESRI^TM^). ArcGIS Model Builder [65] was used to automate the process and Microsoft Office Excel 2016 was used to summarize the results.

### 2.9. Validation of Sun/Shade Model

In order to assess the validity of the shade/sun model, field visits were conducted to the different parts of the city of Madrid to evaluate the size of building shadows. Three hundred photos were taken at different positions (with exact coordinates) in the study area. The photos were taken at different times of the day in the summer of 2018 to represent building shades. Appendix B shows the validation procedure.

As a means of validating the proposed WI, in Table 1 a comparison was made between the WI factors considered in this study and those used for other WIs in previous studies.

## 3. Results

Questionnaire results are shown in Table 3. Values correspond to the average weights awarded to each factor by respondents in the range [0–10].

A WI score was assigned to each of Madrid’s sidewalk polygons at different times. Figure 2 shows a detail extracted from the four maps showing score results for summer/winter and morning/afternoon. The area covered was the old quarter (*centro histórico*) in the center of the city. Variations of the WI sections on the footpath through the different considered times were illustrated by natural breaks in five groups from the lowest to highest values. The five ranges were [0, 21], [21, 32], [32, 42], [42, 52], [52, 85] obtained from the natural break distribution of WI value on summer solstice day at 11:00. Appendix C shows the four maps (Figure A2, Figure A3, Figure A4 and Figure A5) of the entire city and the central part at a larger scale.

The results shown in Window 4 (winter solstice at 17:00) showed fewer variations, as all sidewalks were covered by shade. The values at 13:00 on the winter solstice (Window 3) were more diverse given that some sidewalks received direct sunlight at that time. A similar situation can be observed in Windows 1 and 2 (summer solstice). This demonstrates that the model correctly represents the effect of the changing position of the sunshade for each section of the sidewalk.

It is worth noting that the wide sidewalk on the eastern side had a relatively low WI. This was due to the fact that the tree shade was not computed in the model. A natural evolution of the index would be to include a database of trees and street furniture and their projected shadows.

Throughout the city, the minimum value of the WI at all measured times was 1. The maximum value at both times on the summer day and at 13:00 on the winter day affected by the shade factor was 85, and the maximum value of the WI at 17:00 on the winter day was 66.

The WI was calculated over 32.1 km^2^ of sidewalks. Each section of sidewalk had its own unique WI value. Figure 3 shows the percentages of sidewalks in each of the different ranges of walkability and the percentages of shaded areas for the 21 districts in Madrid.

Figure 3 shows that most of Madrid’s sidewalks have relatively low WI values in the afternoon of the winter solstice as all sidewalk sections were in shade. This makes the high and highest ranges of the WI almost disappear. This explains why red was the dominant color in Map 4 (Figure A5) of Appendix C.

A comparison between Diagrams 3 and 4 showed the effect of the high percentages of sunny sidewalks, which produced higher WI values at 13:00 on the winter solstice, which made the total set of WI values more diverse.

Likewise, if we compare Diagrams 1 and 2 with Diagram 4 we can see that relatively small percentages of shade can cause shifts from low and moderate to high and highest WI ranges. Diagrams 1 and 2 (summer solstice) were very similar. This was due to the fact that most areas were equally sunny at the two different times, even though there was a different WI on either side of the street as shown in Figure 2.

In what regards the effect of the noise factor over the WI values, we tested and computed our index under two different conditions; prior to recent traffic restrictions implemented by Madrid city council in the city center [66] and after these restrictions had been put in place. Measures implemented under this initiative include promoting walking, cycling, and public transportation modes over private cars, enlargement of sidewalks, and banning of non-residents cars in the area. Results are shown in Figure 4. Maps in this figure show WI changes over the street network. Pie charts, in turn, show how the “very low” walkable areas have decreased from 26% to 2%, whereas “high” and “very high” walkable areas have increased from 33% to 54%.

## 4. Discussion

This study produced a detailed WI at sidewalk level. This WI was thematically rich and took dynamic factors related to comfort into account. Previous studies used a static WI and generally took census sections as the spatial observation unit [22,40]. The novelty and value added to the index proposed in this paper lies in the fact that it changes over time as it takes into account daily and seasonal fluctuations in the noise and shade variables.

The proposed WI fills a gap in the available methods for estimating intra-urban walkability and facilitates the computation of spatially detailed WI without the need for field work in any city possessing an up-to-date Spatial Data Infrastructure (SDI). WI results, in this case, are computed at specific small areas. Although noise maps are not part of the core SDI datasets, at the European Union and other developed countries, production of noise maps every five years in cities over 100,000 inhabitants are compulsory by law [67]. Core SDI in the European Union does not include a data layer related to sun and shade. However, EU SDI does include cadaster maps from which elevation and partial data about shade can be computed. Cadaster maps describe parcel level characteristics of plot size and building height in European cities and can be used to calculate sun and shade patterns. Better data are needed concerning tree cover and transient aspects of the environment such as canopies.

The questionnaires completed in relation to the individual weight of each of the WI factors can provide significant information for subsequent studies on walkability. Previous WI proposals [19,30], which weighted all factors equally could now be revised and perhaps improved.

Counting on a spatially detailed WI such as the one we are proposing has the potential to greatly assist decision makers to implement measures promoting walking and, therefore, healthier population habits. Since the WI covers the entire city, any area in need of intervention can be tackled. Intervention examples to alleviate the heat under sunny areas over summer include the installation of street canopies (Appendix D). In regards to the reduction of noise to increase walking, authorities can use the index to implement traffic restriction measures. These are only some examples of the potential use in public health promotion of the WI proposed. In addition to this, as opposed to previous WI, the one we propose allows decision makers to implement dynamic policies according to specific time-related needs.

Limitations of this study are summarized as follows:

Data quality can always be an issue. In the case of Madrid, we detected some missing or wrong data on sidewalks. Unfortunately, there was no data available on trees or street furniture, which prevented more accurate modelling of shady areas. We received 66 (25%) responses out of the 266 questionnaire invitations sent. A larger number of responses would have contributed to strengthening our conclusions. However, considering the very limited number of researchers worldwide looking at this issue, we consider 66 responses as more than acceptable.

Validation of WI remains an issue as walkability scores are not necessarily related to the number of people found walking. Since the novel factors of the WI proposed here are noise and shade/sun conditions, we have conducted experimental fieldwork to verify if people walk in the shady areas on hot days and inversely on sunny areas in winter. Results support our claim (see Appendix D).

## 5. Conclusions

In this study we researched the effects of comfort factors on WI, coming to the conclusion that comfort factors can cause significant changes in the WI in the city of Madrid measured at different times of the day and during different seasons (summer and winter). The distance from busy, noisy streets is another important factor. As noise cartographic products show, noise levels decrease as we get away from busy roads. This study also demonstrates that the available official secondary data sources for Madrid can produce useful, inexpensive, quickly calculated, and detailed WI. Besides, the relative and individual weights of WI factors were identified after considering new comfort factors by sending questionnaires to well-known experts in the field.

## Figures and Tables

**Figure 1 ijerph-16-02850-f001:**
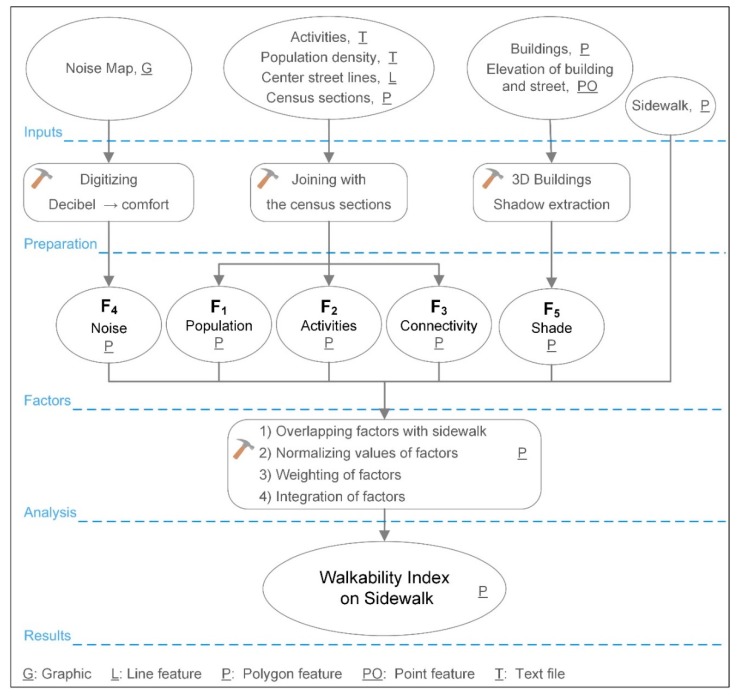
Cartographic model. Input data, technical steps, and results of the WI calculation.

**Figure 2 ijerph-16-02850-f002:**
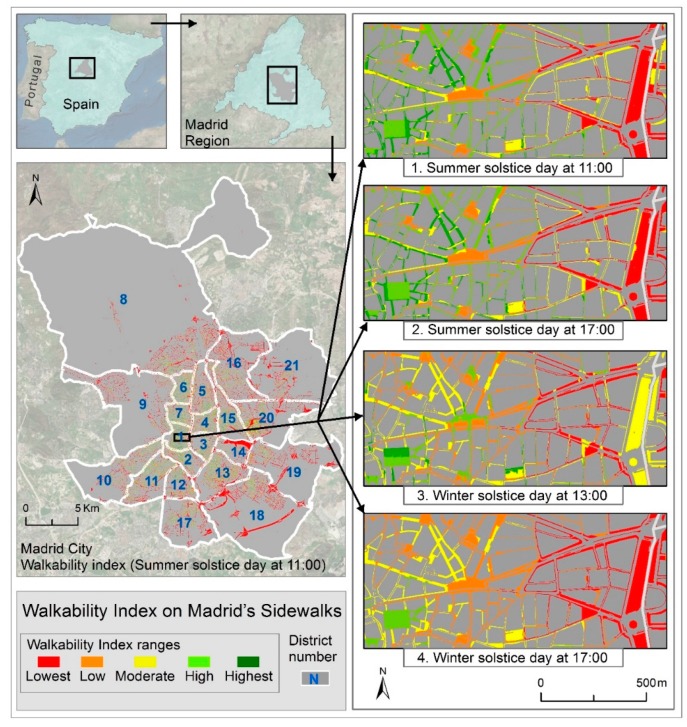
Walkability indices (WI) on a sector of Madrid City at four different times.

**Figure 3 ijerph-16-02850-f003:**
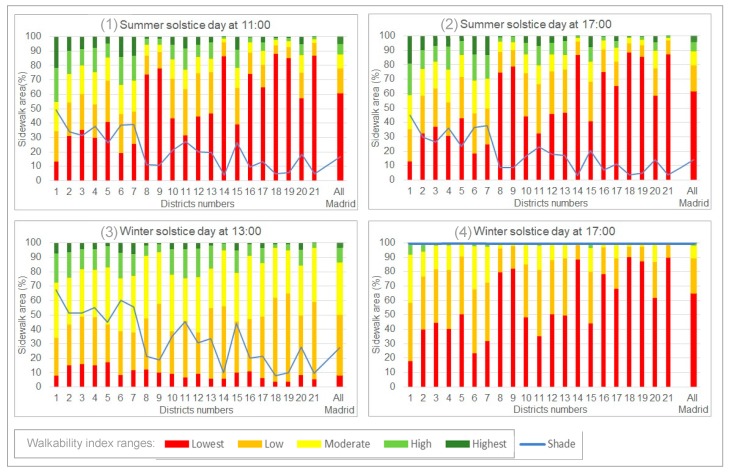
Percentages of WI ranges and shaded areas in each of the 21 districts in Madrid.

**Figure 4 ijerph-16-02850-f004:**
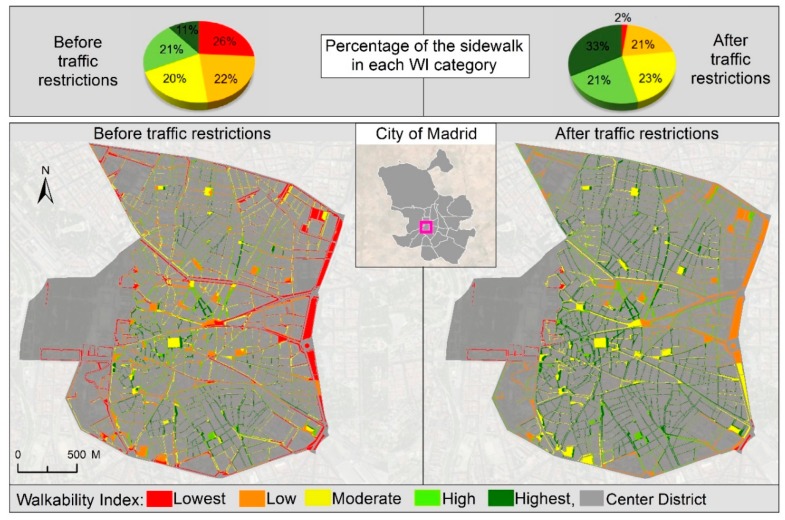
Effect of the noise factor over the proposed WI.

**Table 1 ijerph-16-02850-t001:** Factors and data types in different walkability indices (WI) around the world.

Author	Journal	Country	Walkability Factors					
			Pop. Density	Activities	Connectivity	Noise	Shade	Other
Current study ^S^	-	Spain	yes	yes	yes	yes	Yes	-
2. Habibian et al. (2018) ^S^, [33]	Sustainable Cities and Society	Iran	yes	yes	yes	-	-	destination accessibility
3. Gullón et al. (2017) ^S^, [34]	International Journal of Health Geographics	Spain	yes	-	yes	-	-	Residential Density, Retail Destinations
4. Giles et al. (2014) ^S^, [22]	University of Melbourne	Australia	yes	yes	yes	-	-	-
5. Coffee et al. (2013) ^S^, [35]	Sustainable Cities and Society	Australia	-	yes	yes	-	-	dwelling density and net retail area
6. Freeman et al. (2012) ^S^, [36]	Journal of Urban Health	USA	yes	yes	yes	-	-	subway stop density, the ratio of retail building floor area to retail land area
7. Glazier et al. (2012) ^S^, [37]	ResearchGate	Canada	yes	yes	yes	-	-	dwelling density and availability of retail stores and services within a 10-min walk
9. Frank et al. (2010) ^S^, [21]	British Journal of Sports Medicine	USA	yes	yes	yes	-	-	Retail floor area ratio
10. Van Dyck et al. (2010) ^S^, [38]	Preventive Medicine	Belgian	yes	yes	yes	-	-	-
11. Zhu et al. (2008) ^P,S^ [39]	American Journal of Preventive Medicine	USA	yes	yes	yes	-	tree shade	potential walkers, pedestrian facilities and Neighborhood-level safety Maintenance, Visual quality, Physical amenities, and Safety
12. Leslie et al. (2007) ^S^, [40]	Health and Place	Australia	-	yes	yes	-	-	Dwelling density and Net area retail
8. Pikora et al. (2006) ^P^, [41]	Medicine and Science in Sports and Exercise	Australia	-	-	-	-	-	functional, safety, aesthetic, and destination of physical environmental factors
13. Dannenberg et al. (2004) ^P^, [28]	American Journal of Health Promotion	USA	-	-	-	-	yes	Pedestrian facilities, pedestrian-vehicle conflicts, crosswalks, maintenance, walkway width, roadway buffer, Universal accessibility and aesthetics
14. Walk Score ^S^, [31]	Walk Score^®^	Australia, Canada, USA	yes	yes	yes	-	-	Block length

^P^: Primary data. ^S^: Secondary data.

**Table 2 ijerph-16-02850-t002:** Collected Data.

Data Set	Format	Source	Data Use
Districts, Census sections, Street network center line	Shapefile	National Institute of Statistics (official street map)	Defining factors and presenting the results
Sidewalk		Municipal Cartography (Urban planning and infrastructures) National Land Registry	Capturing all values of the walkability index
Buildings, Land Parcels, Elevation of streets and buildings			Shade factor
Strategic noise map	Image	General Directorate of Sustainability and Environmental Control (Environment and Mobility)	Noise factor
Type of business activities	Text file	Activities census (Economy)	Diversity of activities factor
Demographic Census		Municipal Register (Demography)	Population density factor

**Table 3 ijerph-16-02850-t003:** Factor weights of WI and their individual importance.

Walkability Index Factors	Population Density	Diversity of Activities	Connectivity	Absence of Noise	Shade Factor
Mean individual importance [0–10]	6.65	7.94	7.94	5.82	6.80
Factor weights	WF1 = 0.19	WF2 = 0.23	WF3 = 0.23	WF4 = 0.16	W5 = 0.19

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
