# Peer review of "Comfort and Time-Based Walkability Index Design: A GIS-Based Proposal"

_ijerph, 2019, doi:10.3390/ijerph16162850_

Round 1

Reviewer 1 Report

An interesting article proposing a new method and new factors to be included in a walkability index. However, I have a few comments and suggestions:  

I think it is a good idea to add these factors (noise and shade), but are there any articles in the literature that show links between walking and ambient noise and the shade factor? 

Line 119: Could it be possible to be more specific about this classification of activities and on what basis they were chosen? Some activities would promote walking more than others. 

Line 144: Would it be possible to have more details on the respondents here. 

Why not put Table 2 in the introduction section and recall it in this section. It is like a review of the literature on walkability indices.  

Line 168: More introduction or discussion elements not in the method section.

The discussion should be more detailed. What are the implications of your research findings for public health policy? Link your results to other research on WI. Table 2 for example. 

Lines 222-223: More details on this aspect are needed. 

In the conclusion: add something like this: the next step is the validation of this index with data on walking practice especially for the two new factors, noise and shade. 

Author Response

The authors would like to express their gratitude to this reviewer for his/her valuable time, thorough revision and very useful comments and suggestions. Thanks to him/her the article has greatly improved.

-          Reviewer 1

I think it is a good idea to add these factors (noise and shade), but are there any articles in the literature that show links between walking and ambient noise and the shade factor?

Response: Lines 33-35 include references (25, 26, 27 and 28) to articles that took into account or mentioned the links between walking and these factors. The way they gathered this information is based on subjective measures collected through questionnaires or interviews. The required intensive fieldwork to collect this information made these WI proposals restricted to small urban areas.

In our case, however, these measures are objective, calculated from available sources, and are computed for the whole city (lines 67-72).

Line 119: Could it be possible to be more specific about this classification of activities and on what basis they were chosen? Some activities would promote walking more than others.

Response: We thank reviewer 1 for this comment. We have included a new sentence in lines 125-129.

Line 144: Would it be possible to have more details on the respondents here.

Response: Thank you for this comment. We have added the number of respondents in each group (lines 154-156).

Why not put Table 2 in the introduction section and recall it in this section. It is like a review of the literature on walkability indices. 

Response: We thank the reviewer for this suggestion. We have moved the table to the introduction section (lines 50-51)

Line 168: More introduction or discussion elements not in the method section.

Response: We thank the reviewer for pointing this out. Done (lines 45-49).

The discussion should be more detailed. What are the implications of your research findings for public health policy? Link your results to other research on WI. Table 2 for example.

Response: Thank you for this relevant comment. We have added a new paragraph in lines 236-244

Lines 222-223: More details on this aspect are needed.

Response: Thanks for pointing this out. We have extended our explanation about it in lines 250-253.

In the conclusion: add something like this: the next step is the validation of this index with data on walking practice especially for the two new factors, noise and shade.

Response: We thank you for this relevant comment. We have added a new paragraph in the discussion section (lines 254-258).

Reviewer 2 Report

The paper proposes a walkability index based on a set of variables including activities along the route, population density, street connectivity and measures of sun/shade and noise.
The latter two variables, are seldom considered in the most used measurement methods of walkability, but represent important proxies of street comfort, a quality that affect the propension to walk of citizens. Moreover, they allowed to consider how comfort changes over time.  

The idea of the manuscript is interesting but need to be revised with more attention. 

1) I have some doubts about the formulation of the index, in particular with regard to the selection of variables. 

I agree with the authors when they claimed the necessity of measurement and evaluation methods simple to apply and which result not time and resources consuming. At the same time I believe one of the major opportunities offered by walkability measurement methods is their ability to reveal aspects to be modified by project, being these aspects very often fine-grain characteristics of streetscape (i. e. sidewalk width, crossings, buildings/road relationship, arrangement of the buffer between pedestrian path and carriageway, and so on). The design of the space in-between peds and cars for example, offers many opportunities to improve shade/sun and give protection from traffic noise.  

According to this, the proposed index presents a certain level of methodological contraddiction. From one hand authors look for fast and feasible variables to measure, but from the other hand they decided to include two variables which require an important effort of data processing and validation. This ambiguity need to be clarified.

2) The result section needs to be thoroughly revised. 

In the current version, it only gives comments about the suitabiity and effectiveness of the sun/shade model, and as such it should be switched in section 2.9.

In the discussion of results, instead, the authors should mention the influence of all the variables composing the walkability index. The actual argumentation is restricted to only one analysed variables (sun/shade) thus neglecting the other dimensions some of which (connectivity and land use mixresulted to have a higher importance in the final WI score actually. Noise effect has to be considered as well. 

3) Another consideration about the proposed model concerns the assignation of WI values to spatial features (sidewalk polygons as reported in row 173 and sidewalk section in abstract and in row 189). Authors used very different data, from census to detailed thematic maps (noise maps), and I did not understand clearly the assignation rules. 

Did authors resort to rasterisation? How WI score had been synthesised into a unique sidewalk edges? 

Other observations: 

rows 161-168: the text does not make sense in 2.9 section. It complements the selection of WI variables. I suggest to move it to the introduction. 

Author Response

The authors would like to express their gratitude to this reviewer for his/her valuable time, thorough revision and very useful comments and suggestions. Thanks to him/her the article has greatly improved.

Reviewer 2

I have some doubts about the formulation of the index, in particular with regard to the selection of variables.

I agree with the authors when they claimed the necessity of measurement and evaluation methods simple to apply and which result not time and resources consuming. At the same time I believe one of the major opportunities offered by walkability measurement methods is their ability to reveal aspects to be modified by project, being these aspects very often fine-grain characteristics of streetscape (i. e. sidewalk width, crossings, buildings/road relationship, arrangement of the buffer between pedestrian path and carriageway, and so on). The design of the space in-between peds and cars for example, offers many opportunities to improve shade/sun and give protection from traffic noise.

Response: We are grateful to this reviewer for this thoughtful comment. We absolutely agree. There are some fine-grain, using his/her own words, characteristics of the streetscape that have a great impact on the walkability conditions it offers. We also agree that better designs of the space between pedestrians and cars offer opportunities to improve shade/sun and noise conditions. However, our aim is not to propose a new design but to characterise and score actual conditions so decision makers have a valuable tool to support their decisions to improve urban design. 

According to this, the proposed index presents a certain level of methodological contradiction. From one hand authors look for fast and feasible variables to measure, but from the other hand they decided to include two variables which require an important effort of data processing and validation. This ambiguity need to be clarified.

Response: We respectfully disagree with this comment. Data processing and validation were nothing compared to what it would have been to collect all this information through field work. It would have taken years. It is true that 3d modelling sun/shade conditions is computational intensive but it does not require neither long time or expensive equipment. Lines 130-143 explain this procedure in detail.

The result section needs to be thoroughly revised. In the current version, it only gives comments about the suitability and effectiveness of the sun/shade model, and as such it should be switched in section 2.9.

In the discussion of results, instead, the authors should mention the influence of all the variables composing the walkability index. The actual argumentation is restricted to only one analysed variables (sun/shade) thus neglecting the other dimensions some of which (connectivity and land use mix) resulted to have a higher importance in the final WI score actually. Noise effect has to be considered as well.

Response: Thank you for this comment. Since the proposed WI adds on previous WI two new factors, only those have been commented. We assumed that “classic” factors (diversity of activities, population density, and street connectivity) are sufficiently analysed in the cited literature.

Following this very useful recommendation, we have added a new figure (figure 4) and a new paragraph (lines 213-218) about the effect of the noise factor over the index, complementing this way the information given about the new sun/shade factor.

As what refers to information provided in section 2.9, our purpose is to show that sun/shade calculations were accurate. We do not do the same about noise as this information was provided by the city council and we assumed it is correct. We hope this clarifies reviewer’s concerns.

Another consideration about the proposed model concerns the assignation of WI values to spatial features (sidewalk polygons as reported in row 173 and sidewalk section in abstract and in row 189). Authors used very different data, from census to detailed thematic maps (noise maps), and I did not understand clearly the assignation rules.

Response: We thank this reviewer for this comment and apologise for the lack of clarity. We have added a new line (159-162) to amend it.

Did authors resort to rasterisation? How WI score had been synthesised into a unique sidewalk edges?

Response: No raster format is implied. The sidewalk was partitioned depending on shade/sun conditions. The proposed WI is presented in vector format (polygons). Lines 159-162 add on this.

Rows 161-168: the text does not make sense in 2.9 section. It complements the selection of WI variables. I suggest to move it to the introduction.

Response: Thanks for this comment. We have moved it to the introduction section as suggested (lines 45-51).

Round 2

Reviewer 2 Report

With respect to the first version, the paper has been improved but I ask the authors an additional effort.   

Walkability maps: An explication of how walkability classes have been defined would be useful in order to understand which value of WI has been considered as thresholds. The qualitative classification (low- high) is immediate but it does not help researchers to replicate the tool.

Rows 213-218: Authors wrote "we tested and computed our index under two different conditions; prior to recent traffic restrictions in the city center and after these restrictions have been put in place". 

I do not understand whether the mentioned traffic restrictions are a simulation made by authors or they refer to real measures introduced in the city by traffic authorities? In any case (simulation or description of real life) it would be useful to offer a short descriptions of changes considered. 

what kind of measures did they considered? (traffic calming, pedestrian zones, restricted traffic areas?). a brief discussion of the effects on noise and the WI with comparison to literauture should give more of traffic restrictions on noise in line with  or do they suggest unexpected insights? 

Rows 255-257: The pictures are explanatory of the hot season behaviours (add a photo in winter time to strenghten dimostration). The idea to support the argumentation with photographic materials is good for sun/shade effect, but it does not offer any evidence of noise.  

For this reason I still think the contribution offers innovative cues in the measure of walkability for hat concerns the introduction of sun/shade factors and its computation, while the noise effect is still weak. As I wrote above, a larger argumentation about this point would improve appreciably the paper, expecially with regard to the first assumpions.

For example authors mentioned "the distance from busy, noisy streets" (rows 262-263) as an indicative factor for noise effect. This point could be deepened

Finally a more detailed description of the method used to calculate relative and individual weights of WI factors (subsection 2.6) would complement the paper adding more scientific accuracy and precision.     
